# NICE: NOISE INJECTION AND CLAMPING ESTIMATION FOR NEURAL NETWORK QUANTIZATION

## ABSTRACT

Convolutional Neural Networks (CNN) are very popular in many fields including computer vision, speech recognition, natural language processing, to name a few. Though deep learning leads to ground breaking performance in these domains, the networks used are very demanding computationally and are far from real-time even on a GPU, which is not power efficient and therefore does not suit low power systems such as mobile devices. To overcome this challenge, some solutions have been proposed for quantizing the weights and activations of these networks, which accelerate the runtime significantly. Yet, this acceleration comes at the cost of a larger error. The NICE method proposed in this work trains quantized neural networks by noise injection and a learned clamping, which improve the accuracy. This leads to state-of-the-art results on various regression and classification tasks, e.g., ImageNet classification with architectures such as ResNet-18/34/50 with low as 3-bit weights and 3-bit activations. We implement the proposed solution on an FPGA to demonstrate its applicability for low power real-time applications.

## 1 INTRODUCTION

Deep neural networks have established themselves as an important tool in the machine learning arsenal. They have shown spectacular success in a variety of tasks in a broad range of fields such computer vision, computational and medical imaging, signal, image, speech and language processing (Hinton et al., 2012; Lai et al., 2015; Chen et al., 2018).

However, while deep learning models' performance is impressive, the computational and storage requirements of both training and inference are harsh. For example, ResNet-50 (He et al., 2016), a popular choice for image detection, has 98 MB parameters and requires 4 GFLOPs of computations for a single inference. In many cases, the devices do not have such a big amount of resources, which makes deep learning infeasible in smart phones and the Internet of things (IoT).

In attempt to solve these problems, many researchers have recently came up with less demanding models, often at the expense of more complicated training procedure. Since the training is usually performed on servers with much larger resources, this is usually an acceptable trade-off.

One prominent approach is to quantize the networks. The default choice for the data type of the neural networks' weights and feature maps (activations) is 32-bit (single precision) floating point. Gupta et al. (2015) have shown that quantizing the pre-trained weights to 16-bit fixed point have almost no effect on the accuracy of the networks. Moreover, minor modifications allow performing an integer-only 8-bit inference with reasonable performance degradation (Jacob et al., 2018), which is utilized in DL frameworks, such as TensorFlow. One of the current challenges in network quantization is reducing the precision even further, up to 1-5 bits per value. In this case, straightforward techniques result in unacceptable quality degradation.

**Contribution.** This paper introduces a novel simple approach denoted NICE (noise injection and clamping estimation) for neural network quantization that relies on the following two easy to implement components: (i) Noise injection during training that emulates the quantization noise introduced at inference time; and (ii) Statistics-based initialization of parameter and activation clamping, for faster model convergence. In addition, activation clamp is learned during train time. We also propose integer-only scheme for an FPGA on regression task (Schwartz et al., 2018).

Our proposed strategy for network training lead to an improvement over the state-of-the-art quantization techniques in the performance vs. complexity tradeoff. Unlike several leading methods, our approach can be applied directly to existing architectures without the need to modify them at training (as opposed, for example, to the teacher-student approaches (Polino et al., 2018) that require to train a bigger network, or the XNOR networks (Rastegari et al., 2016) that typically increase the number of parameters by a significant factor in order to meet accuracy goals).

Moreover, our new technique allows quantizing all the parameters in the network to fixed point (integer) values. This include the batch-norm component that is usually not quantized in other works. Thus, our proposed solution makes the integration of neural networks in dedicated hardware devices such as FPGA and ASIC easier. As a proof-of-concept, we present also a case study of such an implementation on hardware. The quantization code will become publicly available upon acceptance.

## 2 RELATED WORK

**Expressiveness based methods.** The quantization of neural network to extremely low-precision representations (up to 2 or 3 possible values) was actively studied in recent years (Rastegari et al., 2016; Hubara et al., 2018; Mishra et al., 2018; Zhang et al., 2018). To overcome the accuracy reduction, some works proposed to use a wider network (Zhu et al., 2016; Polino et al., 2018; Banner et al., 2018), which compensates the expressiveness reduction of the quantized networks network. For example, 32-bit feature maps were regarded as 32 binary ones. Another way to improve expressiveness, adopted by Zhu et al. (2016) and Zhou et al. (2017) is to add a linear scaling layer after each of the quantized layers.

**Keeping full-precision copy of quantized weights.** Lately, the most common approach to training a quantized neural network (Hubara et al., 2016; 2018; Zhou et al., 2016; Rastegari et al., 2016; Cai et al., 2017) is keep two sets of weights — forward pass is performed with quantized weights, and updates are performed on full precision ones, i.e., approximating gradients with straight-through estimator (STE) (Bengio et al., 2013). For quantizing the parameters, either stochastic or deterministic function can be used.

**Distillation.** One of the leading approaches used today for quantization relies on the idea of distillation (Hinton et al., 2015). In distillation a teacher-student setup is used, where the teacher is either the same or a larger full precision neural network and the student is the quantized one. The student network is trained to imitate the output of the teacher network. This strategy is successfully used to boost the performance of existing quantization methods (Mishra & Marr, 2018; Polino et al., 2018; Jung et al., 2018).

**Model parametrization.** Zhang et al. (2018) proposed to represent the parameters with learned basis vectors that allow acquiring an optimized non-uniform representation. In this case MAC operations can be computed with bitwise operations. Choi et al. (2018) proposed to learn the clamping value of the activations to find the balance between clamping and quantization errors. In this work we also learn this value but with the difference that we are learning the clamps value directly using STE back-propagation method without any regulations on the loss Jung et al. (2018) created a more complex parametrization of both weights and activations, and approximated them with symmetric piecewise linear function, learning both the domains and the parameters directly from the loss function of the network.

**Optimization techniques.** Zhou et al. (2017) and Dong et al. (2017) used the idea of not quantizing all the weights simultaneously but rather gradually increasing the number of quantized weights to improve the convergence. McKinstry et al. (2018) demonstrated that 4-bit fully integer neural networks can achieve full-precision performance by applying simple techniques to combat varience of gradients: larger batches and proper learning rate annealing with longer training time. However, 8-bit and 32-bit integer representations were used for the multiplicative (i.e., batch normalization) and additive constants (biases), respectively.

**Generalization bounds.** Interestingly, quantization of neural networks have been used recently as a theoretical tool to understand better the generalization of neural networks. It has been shown that while the generalization error does not scale with the number of parameters in over-parameterized networks, it does so when these networks are being quantized (Arora et al., 2018).

## 3 METHOD

In this work we propose a training scheme for quantized neural networks designed for fast inference on hardware with integer-only arithmetic. To achieve maximum performance, we apply a combination of several well-known as well as novel techniques. Firstly, in order to emulate the effect of quantization, we inject additive random noise into the network weights. Uniform noise distribution is known to approximate well the quantization error for fine quantizers; however, our experiments detailed in the sequel show that it is also suitable for relatively coarse quantization (Appendix A). Furthermore, some amount of random weight perturbation seems to have a regularization effect beneficial for the overall convergence of the training algorithm. Secondly, we use a gradual training scheme to minimize the perturbation of network parameters performed simultaneously. In order to give the quantized layers as much gradient updates as possible, we used the STE approach to pass the gradients to the quantized layers. After the gradual phase, the whole network is quantized and trained for a number of fine-tuning epochs. Thirdly, we propose to clamp both the activations and the weights in order to reduce the quantization bin size (and, thus, the quantization error) at the expense of some sacrifice of the dynamic range. The clamping values are initialized using the statistics of each layer. In order to truly optimize the tradeoff between the reduction of the quantization error vs that of the dynamic range, we learn optimal clamping values by defining a loss on the quantization error.

Lastly, following common we don't quantize first and last layers of the networks, The remainder of the section details these main ingredients of our method.

We propose to inject uniform additive noise to weights and biases during model training to emulate the effect of quantization incurred at inference. Prior works have investigated the behavior of quantization error (Sripad & Snyder, 1977; Gray, 1990) and concluded that in sufficiently fine-grain quantizers it can be approximated as a uniform random variale. We have observed the same phenomena and empirically verified it for weight quantization as coarse as $5$ bits.

The advantage of the proposed method is that the updates performed during the backward pass immediately influence the forward pass, in contrast to strategies that directly quantize the weights, where small updates often leave them in the same bin, thus, effectively unchanged.

In order to achieve a dropout-like effect in the noise injection, we use a Bernoulli distributed mask $M$, quantizing part of the weights and adding noise to the others. From empirical evidence, we chose $M \sim \text{Ber}(0.05)$ as it gave the best results for the range of bitwidths in our experiments. Instead of using the quantized value $\hat{w} = \mathcal{Q}_\Delta(w)$ of a weight $w$ in the forward pass, $\hat{w} = (1 - M)\mathcal{Q}_\Delta(w) + M(w - e)$ is used with $e \sim \text{Uni}(-\Delta/2, \Delta/2)$, where $\Delta$ denotes size of the quantization bin.

### 3.1 GRADUAL QUANTIZATION

In order to improve the scalability of the method for deeper networks, it is desirale to avoid the significant change of the network behavior due to quantization. Thus, we start from gradually adding a subset of weights to the set of quantized parameters, allowing the rest of the network to adapt to the changes.

The gradual quantization is performed in the following way: the network is split into $N$ equally-sized blocks of layers $\{B_1, ..., B_N\}$. At the $i$-th stage, we inject the noise into the weights of the layers from the block $B_i$. The previous blocks $\{B_1, ..., B_{i-1}\}$ are quantized, while the following blocks $\{B_{i+1}, ..., B_N\}$ remain at full precision. We apply the gradual process only once, i.e., when the $N$-th stage finishes, in the remaining training epochs we quantize and train all the layers using the STE approach.

This gradual increasing of the number of quantized layers is similar to the one proposed by Xu et al. (2018). This gradual process reduces, via the number of parameters, the amount of simultaneously injected noise and improves convergence. Since we start from the earlier blocks, the later ones have an opportunity adapt to the quantization error affecting their inputs and thus the network does not change drastically during any phase of quantization. After finishing the training with the noise injection into the block of layers $B_N$, we continue the training of the fully quantized network for several epochs until convergence. In the case of a pre-trained network destined for quantization, we have found that the optimal block size is a single layer with the corresponding activation, while using more than one epoch of training with the noise injection per block does not improve performance.

### 3.2 CLAMPING AND QUANTIZATION

In order to quantize the network weights, we clamp their values in the range $[-c_w, c_w]$:

$$w_c = \text{Clamp}(w, -c_w, c_w) = \max\left(-c_w, \min\left(x, c_w\right)\right). \tag{1}$$

The parameter $c_w$ is defined per layer and is initialized with $c_w = \text{mean}(w) + \beta \times \text{std}(w)$, where $w$ are the weighs of the layer and $\beta$ is a hyper-parameter. Given $c_w$, we uniformly quantize the clamped weight into $B_w$ bits according to

$$\hat{w} = \left[ w_c \frac{2^{B_w-1} - 1}{c_w} \right] \frac{c_w}{2^{B_w-1} - 1},$$

where $[\cdot]$ denotes the rounding operation.

The quantization of the network activations is performed in a similar manner. The conventional ReLU activation function in CNNs is replaced by the clamped ReLU,

$$a_c = \text{Clamp}(a, 0, c_a), \tag{2}$$

where $a$ denotes the output of the linear part of the layer, $a_c$ is nonnegative value of the clamped activation prior to quantization, and $c_a$ is the clamping range. The constant $c_a$ is set as a local parameter of each layer and is learned with the other parameters of the network via backpropagation. We used the initialization $c_a = \text{mean}(a) + \alpha \times \text{std}(a)$ with the statistics computed on the training dataset and $\alpha$ set as a hyper-parameter.

A quantized version of the truncated activation is obtained by quantizing $a_c$ uniformly to $B_a$ bits,

$$\hat{a} = \left[ a_c \frac{2^{B_a} - 1}{c_a} \right] \cdot \frac{c_a}{2^{B_a} - 1}. \tag{3}$$

Since the Round function is non-differentiable, we use the STE approach to propagate the gradients through it to the next layer. For the update of $c_a$, we calculate the derivative of $\hat{a}$ with respect to $c_a$ as

$$\frac{\partial \hat{a}}{\partial a_c} = \begin{cases} 1, & a_c \in [0, c_a] \\ 0, & \text{otherwise}. \end{cases} \tag{4}$$

Additional analysis of the clamping parameter convergence is presented in Appendix B.

The quantization of the layer biases is more complex, since their scale depends on the scales of both the activations and the weights. For each layer, we initialize the bias clamping value as

$$c_b = \left( \underbrace{\frac{c_a}{2^{B_a} - 1}}_{\text{Activation scale}} \cdot \underbrace{\frac{c_w}{2^{B_w - 1} - 1}}_{\text{Weight scale}} \right) \cdot \left( \underbrace{2^{B_b - 1} - 1}_{\text{Maximal bias value}} \right), \tag{5}$$

where $B_b$ denotes the bias bitwidth. The biases are clamped and quantized in the same manner as the weights.

## 4 EXPERIMENTAL RESULTS

To demonstrate the effectiveness of our method, we implemented it in PyTorch and evaluated on image classification datasets (ImageNet and CIFAR-10) and a regression scenario (the MSR joint denoising and demosaicing dataset (Khashabi et al., 2014)). The CIFAR-10 results are presented in Appendix C. In all the experiments, we use a pre-trained FP32 model, which is then quantized using NICE .

### 4.1 IMAGENET

For quantizing the ResNet-18/34/50 networks for ImageNet, we fine-tune a given pre-trained network using NICE . We train a network for a total of 120 epochs, following the gradual process described in Section 3.1 with the number of stages $N$ set to the number of trainable layers. We use an SGD optimizer with learning rate is $10^{-4}$, momentum 0.9 and weight decay $4 \times 10^{-5}$.

Table 2 compares NICE with other leading approaches to low-precision quantization (Jung et al., 2018; Choi et al., 2018; Zhang et al., 2018; McKinstry et al., 2018). Various quantization levels of the weights and activations are presented. As a baseline, we use a pre-trained full-precision model.

Our approach achieves state-of-the-art results for 4 and 5 bits quantization and comparable results for 3 bits quantization, on the different network architectures. Moreover, notice that our results for the 5,5 setup, on all the tested architectures, have slightly outperformed the FAQ 8,8 results.

### 4.2 REGRESSION - JOINT DENOISING AND DEMOSAICING

Table 1: PSNR [dB] results on joint denoising and demosaicing for different bitwidths.

| Method | Bits (w=32,a=32) | Bits (w=4,a=8) | Bits (w=4,a=6) | Bits (w=4,a=5) | Bits (w=3,a=6) |
|---|---|---|---|---|---|
| NICE (Ours) | 39.696 | 39.456 | 39.332 | 39.167 | 38.973 |
| WRPN (our experiments) | 39.696 | 38.086 | 37.496 | 36.258 | 36.002 |

In addition to the classification tasks, we apply NICE on a regression task, namely joint image denoising and demosaicing. The network we use is the one proposed in (Schwartz et al., 2018). We slightly modify it by adding to it Dropout with $p = 0.05$, removing the tanh activations and adding skip connections between the input and the output images. These skip connections improve the quantization results as in this case the network only needs to learn the necessary modifications to the input image. Figure 1 shows the whole network, where the modifications are marked in red. The three channels of the input image are quantized to 16 bit, while the output of each convolution, when followed by an activation, are quantized to 8 bits (marked in Fig. 1). The first and last layers are also quantized.

We apply NICE on a full precision pre-trained network for 500 epochs with Adam optimizer with learning rate of $3 \cdot 10^{-5}$. The data is augmented with random horizontal and vertical flipping. Since we are not aware of any other work of quantization for this task, we implemented WRPN (Mishra et al., 2018) as a baseline for comparison. Table 1 reports the test set PSNR for the MSR dataset (Khashabi et al., 2014). It can be clearly seen that NICE achieves significantly better results than WRPN, especially for low weight bitwidths.

### 4.3 ABLATION STUDY

In order to show the importance of each part of our NICE method, we use ResNet-18 on ImageNet. Table 3 reports the accuracy for various combinations of the NICE components. Notice that for high bitwidths, i.e., 5,5 the noise addition and gradual training contribute to the accuracy more than the clamp learning. This happens since (i) the noise distribution is indeed uniform in this case as we show

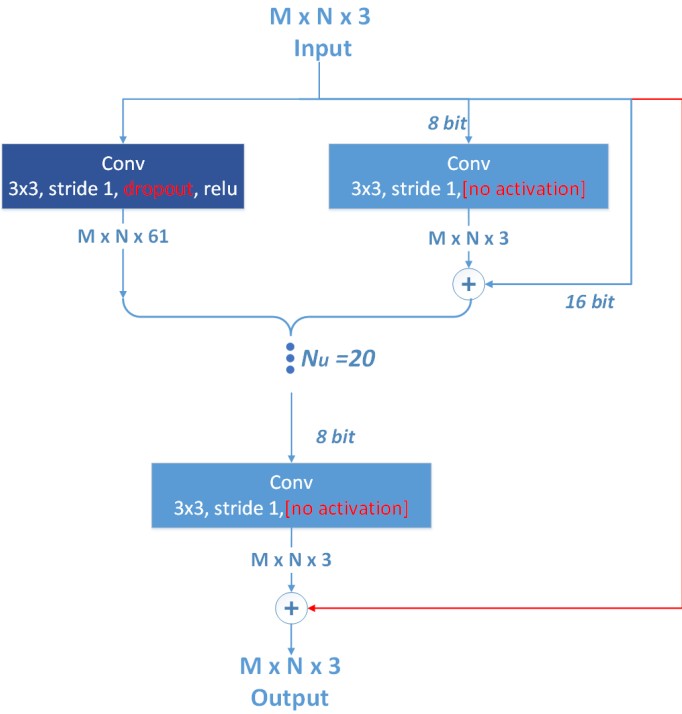

Figure 1: Model used in denoising/demosaicing experiment

in Appendix A; and (ii) the relatively high number of activation quantization levels almost negates the effect of clamping. For low bitwidths, i.e 3,3, we observe the opposite. The uniform noise assumption is no longer accurate. Moreover, due to the small number of bits, clamping the range of values becomes more significant.

## 5 HARDWARE IMPLEMENTATION

### 5.1 OPTIMIZING QUANTIZATION FLOW FOR HARDWARE INFERENCE

Our quantization scheme can fits an FPGA implementation well for several reasons. Firstly, uniform quantization of both the weights and activation induces uniform steps between each quantized bin. This means that we can avoid the use of a resource costly code-book (look-up table) with the size $B_a \times B_w \times B_a$, for each layer. This also saves calculation time.

Secondly, our method enables having an integer-only arithmetic. In order to achieve that, we first represent both activations and parameters as $X = N \times S$, where N is the integer code and S are the pre-calculated scales as can be seen in Equation equation 5 . We then replace the scaling factors $S$ to the form $\hat{S} = q \times 2^p$ where $q \in \mathbb{N}, p \in \mathbb{Z}$. Practically, we found that its sufficient to constrain these values to $q \in [1, 256], p \in [-32, 0]$ without accuracy drop .This representation allows the replacement of hardware costly floating point operations by a combination of cheap shift operation and integer arithmetic.

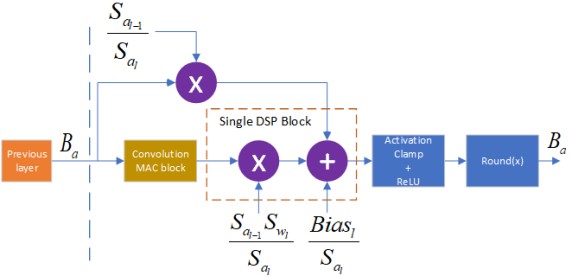

Figure 2: Residual block in hardware

## 5.2 HARDWARE FLOW

In this work, for both regression and classification tasks, we adopt PipeCNN implementation released by the authors.[1] In this implementation, the FPGA is programmed with an image containing: data moving, convolution and a pooling kernels. Layers are calculated sequentially. Figure 2 illustrates the flow of feature maps in residual block from previous layer to the next one. $Sa_i, Sw_i$ are activation and weights scale factors of layer $i$, respectively. All the scaling factors are calculated off-line and are loaded to the memory along with the rest of the parameters. In this paper, FPGA is used for inference only. We have compiled the OpenCL kernel to Intel's Arria 10 FPGA and run it with the DeepISP architecture. Weights were quantized to 4 bits activations to 8 bits, biases and the input image to 16 bits. Resource utilization amounted to 222K LUTs, 650 DSP Blocks and 35.3 Mb of on-chip RAM. With the maximum clock frequency of 240MHz, the processing of a single image took 250ms. In terms of the energy envelope, computation on the FPGA was over 20% more efficient than an equivalent computation on an NVIDIA Titan X GPU. From standard harware design practices, we can project that a dedicated ASIC manufactured using a similar process would be more efficient by at least one order of magnitude.

## 6 CONCLUSION

We introduced NICE – a training scheme for quantized neural networks. The scheme is based on using uniform quantized parameters, additive uniform noise injection and learning quantization clamping range. The scheme is amenable to efficient training by back propagation in full precision arithmetic.

We reported state-of-the-art results on ImageNet for a range different bitwidths and network architectures. Our solution outperforms current works on both 4,4 and 5,5 setups, for all tested architectures, including non-uniform solutions such as (Zhang et al., 2018). It shows comparable results for 3,3 setup.

We showed that quantization error for 4 and 5 bits distributes uniformly, which is why additive uniform noise improved the results more to these bitwidths, than 3 bits. This implies that the results for 3 bits can be furthered improved by adding non-uniform noise to the parameters. The 4 bits setup is of special interest since it is considered more hardware friendly, and due to the upcoming release of Nvidia's Turing architecture which contains INT4 tensor cores.

NICE is straightforward to implement and can be used as a "plug-and-play" modification of existing architectures. It also does not require tweaking the architecture, e.g. increasing the number of filters as done in few previous works.

---

[1]https://github.com/doonny/PipeCNN

Table 2: ImageNet comparison. We report top-1, top-5 accuracy on ImageNet compared with state-of-the-art prior methods. For each DNN architecture, rows are sorted in number of bits.Baseline results were token from PyTorch model zoo. Compared methods: JOINT (Jung et al., 2018), PACT (Choi et al., 2018), LQ-Nets (Zhang et al., 2018), FAQ (McKinstry et al., 2018)

| Network | Method | Precision (w,a) | Accuracy (% top-1) | Accuracy (% top-5) |
|---------|--------|-----------------|--------------------|--------------------|
| ResNet-18 | baseline | 32,32 | 69.76 | 89.08 |
| ResNet-18 | FAQ | 8,8 | 70.02 | 89.32 |
| ResNet-18 | NICE (Ours) | 5,5 | **70.35** | **89.8** |
| ResNet-18 | PACT | 5,5 | 69.8 | 89.3 |
| ResNet-18 | NICE (Ours) | 4,4 | **69.79** | **89.21** |
| ResNet-18 | JOINT | 4,4 | 69.3 | - |
| ResNet-18 | PACT | 4,4 | 69.2 | 89.0 |
| ResNet-18 | FAQ | 4,4 | **69.81** | 89.10 |
| ResNet-18 | LQ-Nets | 4,4 | 69.3 | 88.8 |
| ResNet-18 | JOINT | 3,3 | **68.2** | - |
| ResNet-18 | NICE (Ours) | 3,3 | 67.68 | **88.2** |
| ResNet-18 | LQ-Nets | 3,3 | **68.2** | 87.9 |
| ResNet-18 | PACT | 3,3 | 68.1 | **88.2** |
| ResNet-34 | baseline | 32,32 | 73.30 | 91.42 |
| ResNet-34 | FAQ | 8,8 | 73.71 | **91.63** |
| ResNet-34 | NICE (Ours) | 5,5 | **73.72** | **91.60** |
| ResNet-34 | NICE (Ours) | 4,4 | **73.45** | **91.41** |
| ResNet-34 | FAQ | 4,4 | 73.31 | 91.32 |
| ResNet-34 | LQ-Nets | 3,3 | **71.9** | 88.15 |
| ResNet-34 | NICE (Ours) | 3,3 | 71.74 | **90.8** |
| ResNet-50 | baseline | 32,32 | 76.15 | 92.87 |
| ResNet-50 | FAQ | 8,8 | 76.52 | 93.09 |
| ResNet-50 | PACT | 5,5 | 76.7 | 93.3 |
| ResNet-50 | NICE (Ours) | 5,5 | **76.73** | **93.31** |
| ResNet-50 | NICE (Ours) | 4,4 | **76.5** | **93.3** |
| ResNet-50 | LQ-Nets | 4,4 | 75.1 | 92.4 |
| ResNet-50 | PACT | 4,4 | **76.5** | 93.2 |
| ResNet-50 | FAQ | 4,4 | 76.27 | 92.89 |
| ResNet-50 | NICE (Ours) | 3,3 | 75.08 | 92.35 |
| ResNet-50 | PACT | 3,3 | **75.3** | **92.6** |
| ResNet-50 | LQ-Nets | 3,3 | 74.2 | 91.6 |

Table 3: Ablation study of ResNet18 ImageNet Dataset NICE scheme. We measured TOP-1 accuracy

| Noise+Gradual training | Activation clamping learning | Accuracy on 5,5 [W,A] | Accuracy on 3,3 [W,A] |
|------------------------|------------------------------|-----------------------|-----------------------|
| - | - | 69.72 | 66.51 |
| - | ✓ | 69.9 | 67.2 |
| ✓ | - | 70.25 | 66.7 |
| ✓ | ✓ | 70.3 | 67.68 |

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

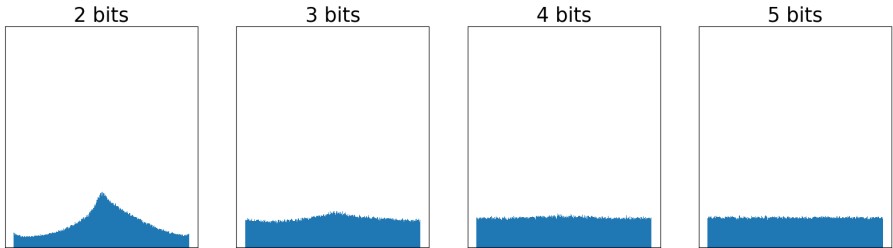

Figure A.1: Weight quantization error histogram for a range of bitwidths

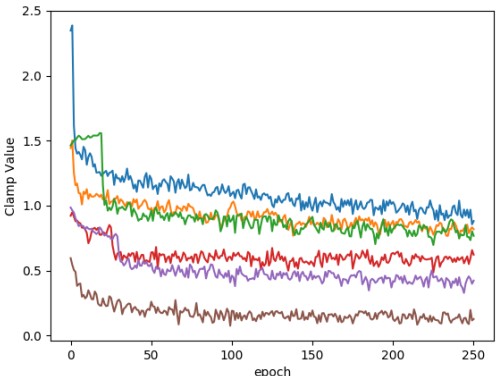

Figure B.1: Activation clamp values during ResNet18 training on CIFAR10 dataset

## A  QUANTIZATION ERROR DISTRIBUTION

The general statement is that for large number of bins, the distribution of quantization error is independent on the quantized value, and thus distributed uniformly. However, this is true only in limit of high number of bins, which is not exactly the case of neural network quantization. However, empirically the distribution of noise is almost uniform for 4 and 5 bits and only starts to deviate deviating from the uniform model (Figure A.1) for 3 bits, which corresponds to only 8 bins.

## B  CLAMPING PARAMETER CONVERGENCE

Figure B.1 depicts the evolution of the activation clamp values throughout the epochs. In this experiment $\alpha$ was set to 5. It can be seen that activation clamp values converge to values smaller than the initialization. This shows that the layer prefers to shrink the dynamic range of the activations, which can be interpreted as a form of regularization similar in its purpose to weight decay on weights.

## C  EXPERIMENTS ON CIFAR-10

As an additional experiment, we test NICE with ResNet-18 on CIFAR-10 for various quantization levels of the weights and activations. Table C.1 reports the results. Notice that for the case of 3-bit weights activations

we get the same accuracy and for the 2-bit case only a small degradation. Moreover, observe that when we quantize only the weights or activations, we get a nice regularization effect that improves the achieved accuracy.

Table C.1: NICE Accuracy (% top-1) on CIFAR-10 for range of bitwidths.

|  |  | Activation bits | | | |
|---|---|---|---|---|---|
|  |  | 1 | 2 | 3 | 32 |
| Weight bits | 2 | 89.5 | 92.53 | 92.69 | 92.71 |
|  | 3 | 91.32 | 92.74 | 93.01 | 93.26 |
|  | 32 | 91.87 | 93.04 | 93.15 | 93.02 |

# D   BACKGROUND FOR NEURAL NETWORKS ON CUSTOM HARDWARE

When implementing systems involving arbitrary precision, FPGAs and ASICs are a natural selection as target device due to their customizable nature. It was already shown that there is a lot of redundancy when using floating point representation in Neural Network(NN). Therefore, custom low-precision representation can be used with little impact to the accuracy. Due to the steadily increasing on-chip memory size (tens of megabytes) and the integration of high bandwidth memory (hundreds of megabytes), it is feasible to fit all the parameters inside an ASIC or FPGA, when using low bitwidth. Besides the obvious advantage of reducing the latency, this approach has several advantages: power consumption reduction and smaller resource utilization, which in addition to DSP blocks and LUTs, also includes routing resource. The motivation of quantizing the activations is similar to that of the parameters. Although activations are not stored during inference, their quantization can lead to major saving in routing resources which in turn can increase the maximal operational frequency of the fabric, resulting in increased throughput. In recent years, FPGAs has become more popular as an inference accelerator. And while ASICs (Chen et al., 2016; Jouppi et al., 2017) usually offers more throughput with lower energy consumption, they don't enjoy the advantage of reconfigurability as FPGAs. This is important since neural network algorithm evolve with time, so should their hardware implementation. Since the implementation of neural network involves complex scheduling and data movement, FPGA-based inference accelerators has been described as heterogeneous system using OpenCL (Wang et al., 2016a; Aydonat et al., 2017; Wang et al., 2016b) or as standalone accelerator using HLS compilers (Umuroglu et al., 2017; Zhao et al., 2017; Ghaffari & Sharifian, 2016)

