# OpenReview forum: "NICE: noise injection and clamping estimation for neural network quantization"
_ICLR.cc/2019/Conference_

### Official Review · AnonReviewer3 · 2018-10-27
**promising results but presentation needs some work**

**Rating:** 4
**Confidence:** 3

**Review:**

The authors present a method for fine-tuning neural networks so inference can be performed in a quantized low bit data format down to 3 bits. The authors achieve this through a combination of three techniques:
1. Noise injection to fine-tune the weights before quantization. The effect of noise injection can model that of quantization, but rather than being stuck in a quantization bin, fine grained weight updates are still possible
2. A schedule that quantizes layer by layer, rather than all layers at the same time
3. Clipping weights and activations within a learned range to obtain finer grained bins within that range.


The main contribution is a novel combination of mostly existing techniques. Clipping (or clamping as the authors call it) has been proposed by Zhang et al. 2018, but it's an interesting contribution to have the clipping learned directly via backpropagation with a straight-through estimator. Treating the quantization as noise has been proposed in a different form in McKinstry et al. 2018. Gradual quantization appears novel, but is also the least interesting of the techniques. Therefore, novelty on ideas/methods is somewhat limited, and the contribution is mostly the in the impressive experimental results, which appear to be outperforming previous methods. The main weaknesses are poor writing, and that some details of the implementation required to reproduce the results are missing. For example, the training schedule is not given, e.g. how many epochs to train the clean model, how many with noise, how many quantized. Details on the gradual quantization are also missing. Block based quantization is completely heuristic and not well motivated. If this is the main novel ingredient, more details on the mechanics would be needed. Is both the noise injection and the quantization done in blocks? If the motivation is in "the opportunity to adapt”, then what does the adaptation look like?

As above, my other main issue is with the writing, there are many examples where I would suggest improvements:

This work could be improved greatly by copy editing for English grammar. There are many typos (including ones that can be caught by autocorrect, missing punctuation, or using similar but unrelated words, e.g. "token" instead of "taken"). The manuscript appears hastily put together and not ready for publication.

The acronym NICE already has a meaning in the DL literature: Dinh, L., Krueger, D., & Bengio, Y. (2014). NICE: Non-linear Independent Components Estimation. It confusing to reuse it.

The term clamping is only explained on page 4 but used since the abstract. It’s used in a nonstandard way to mean “constrained to lie within a range” which should be explained earlier. I think “clipped” would be a better term, following the related Choi et al. 2018. Clamping usually means "constrained to a fixed value" (not a range), so it is not a good term to use in this context.

Are the results shown in table 2 and table 3 from a single trial or averaged across reruns? If single trial, it's misleading to have 2 figures after the decimal. Even non-quantized ResNet tends to have 0.5% or so run to run variability, which is much larger than the differences between some of the methods shown here. In fact, a lot of the results could just be due to picking a lucky random seed.

Comparisons are shown against methods JOINT (Jung et al), LQ-Nets (Zhang et al), FAQ (McKinstry et al). It would be helpful to present them with the same names in the "related work" section, and explain why they were picked out for the comparison. For someone not familiar with the literature it's hard to see why these 3 would be the obvious picks.

Readability would increase if table 2 and 3 were moved to section 4 where they are referenced, rather than after the discussion. Fig 2 font size too small and hard to read.

---

### Official Review · AnonReviewer1 · 2018-11-02
**Network quantization on ResNets**

**Rating:** 5
**Confidence:** 3

**Review:**

The article presents a method for quantization of deep neural networks for classification and regression, using three key parts: (i) noise injection to model the effect of quantization during forward inference, (ii) clamping with learned maximum activations to reduce the quantization bin size, and (iii) gradual quantization of blocks of the network, while previously quantized blocks remain unchanged. The method is evaluated on ImageNet, CIFAR-10, and a regression task, showing performance on-par or better than state-of-the-art methods for particular quantization bit size. Finally, the method is used for porting network onto a FPGA.

The paper addresses an important topic, because there is increasing interest in hardware-efficient implementations of deep neural networks. The method could be interesting for practitioners, because it does not interfere with the original training of the full-precision method, and can be applied later on.

The main weakness is that none of the proposed methods are entirely original, and the combination is rather ad-hoc than well-justified. For example, quantization noise has been considered in several previous articles, e.g. already in BinaryConnect (Courbariaux et al. 2015), although the novelty here is that the noise is explicitly added during the forward path. However, the choice of the Bernoulli mask with p=0.05 is not justified and might not work best for other tasks. The authors admit that gradual quantization has been proposed before, and clamping a ReLU is also not new, although here a new way to learn and initialize the clamping parameters is presented.

The article would be OK if the empirical results were really strong, but unfortunately they are not entirely convincing:
1. The classification results are only for ResNet architectures, it remains unclear whether results would hold also for other architectures.
2. The numerical results in Table 2 are very close to each other, and no error bars are available, so it is not possible to judge whether differences are significant. Also, the advantage of the NICE method vanishes for 3-bit models.
3. The results for CIFAR-10 come without any comparison.
4. The results for regression are only compared to a single method, which is re-implemented by the authors, and might therefore not be fully optimized. Thus there is no strong baseline to judge the results.
5. No results are shown for the hardware implementation.

Overall, the paper is not a particularly interesting read for people interested in a deeper understanding of network quantization, but the method could still be valuable for applications. Is this sufficient for ICLR? Since the experimental results do not entirely convince me I will put my grade slightly below acceptance threshold.

Minor points:
- The abstract has a pretty long introduction before it begins to tell what the contributions of the article are.
- Occasional grammar mistakes.
- Tables 1 and 2 are misplaced

Pros:
+ important topic (network quantization)
+ good empirical results
+ easy to apply

Cons:
- combination of previously proposed methods
- no convincing justification
- no strong advantage over previous methods

---

### Official Review · AnonReviewer2 · 2018-11-02
**Low novelty and weak experiments.**

**Rating:** 4
**Confidence:** 4

**Review:**

[Summary]
Neural network quantization is can enable many practical applications for deep learning, therefore it is an important research problem. The paper claims two contributions: 1. Injecting noise during training to make it more robust to quantization errors. 2. Clamping the parameter values in a layer as well as the activation output, where the clamping interval is some multiple of the standard deviation about the mean, and the clamping interval is updated using the Straight through estimator. The main strength of the paper lies in the empirical results where the combination of techniques employed by the authors outperforms the SOTA methods in a compute of scenarios.

[Pros]
The paper is working on an important problem area and as a technical report this work can be valuable in the industry. There is novelty in the particular combination of techniques that the authors have employed and some of the empirical results show the strength of the technique.

[Cons]
the main contribution of the paper is a careful combination of existing techniques and the associated empirical results, therefore the experiments need to be strong. I noticed some strange omissions in the  results, and asked the authors for a reply via a public comment but they did not reply. Specifically,

a) On RESNET 34 the results for PACT 5,5 are not shown and JOINT and PACT on 3,3 on ResNet-34 are also not shown. Why are these results omitted?

b) The noise+gradual training decreases performance on (layer-weight bitwidth, activation bitwidth)  =  (3, 3). But further experiments for table 2 where the nets do not use noise+gradual training is not shown. Currently the proposed recipe for quantizing nets does not seem to be all that better than existing methods and it hard to guess exactly what was the reason for the improved results in situations where the results were infact better. Why were these experiments omitted ?

Overall the experimental results in the paper are weak and the novelty of the proposed methods is low.

---

### Comment · AnonReviewer2 · 2018-10-25
**On omitted experiments and citations**

This paper is working on an important problem. However, since the main contribution of the paper is a careful combination of existing techniques and the associated empirical results, therefore the experiments need to be strong. I have noticed some strange omissions in the  results.

a) On RESNET 34 the results for PACT 5,5 are not shown and JOINT and PACT on 3,3 on ResNet-34 are also not shown. Why are these results omitted?

b) The noise+gradual training decreases performance on (layer-weight bitwidth, activation bitwidth)  =  (3, 3). But further experiments for table 2 where the nets do not use noise+gradual training is not shown. Currently the proposed recipe for quantizing nets does not seem to be all that better than existing methods and it hard to guess exactly what was the reason for the improved results in situations where the results were infact better. Why were these experiments omitted ?

c) You mention in the paper that "following common [sic] we don’t quantize first and last layers of the networks" What is the citation for this claim? The paper by Arora et. al. that you do cite in the paper says the exact opposite. Their experiments on compression shows that later layers  (including the last layer) are more compressible than the earlier layers.

---

### Meta-Review · Area_Chair1 · 2018-12-13
**Addresses an important problem, but the novelty is limited and experiments may not be good enough**

**Confidence:** 5
**Recommendation:** Reject

**Metareview:**

This paper addresses an important problem, quantizing deep neural network models to reduce the cost of implementing them on hardware such as FPGAs without severely affecting task performance. The approach explored in the paper combines three ideas: (1) injecting noise into the network to simulate the effects of quantization noise, (2) a smart initialization of the parameter and activation clamping along with learning of the activation clamping using the straight-through estimator, and (3) a gradual approach to quantization. While the reviewers agreed that the problem is important, they raised concerns about the novelty of the proposed approach and the quality of the experiments. The authors did not respond to the reviewers in the discussion period, and did not revise their submission.